# Calculation of Parasitic Capacitance to Analyze Shaft Voltage of Electric Motor with Direct-Oil-Cooling System

Chan-Ho Kim [1] , Sung-Bae Jun [1] , Han-Joon Yoon [1], Nam-Ho Kim [1], Ho-Chang Jung [2], Rae-Eun Kim [3] and Sang-Yong Jung [1,*]

1    Department of Electrical and Computer Engineering, Sungkyunkwan University, Suwon 16419, Korea
2    Advanced Powertrain R&D Center, Korea Automotive Technology Institute, Cheonan 31214, Korea
3    Intelligent Mechatronics Research Center, Korea Electronics Technology Institute, Bucheon 14502, Korea
*    Correspondence: syjung@skku.edu

**Abstract:** In modern electric vehicles, electrical failure has become a critical problem that reduces the lifetime of traction motors. Moreover, traction motors with high-voltage and high-speed systems for a high power density have been aggravating the shaft voltage problems. This study identifies that direct-oil-cooling systems exacerbate this problem. To address this, an analytical method for calculating parasitic capacitance is proposed to determine the effects of cooling oil in a traction motor with a direct-oil-cooling system. Capacitance equivalent circuits are configured based on whether the slot is submerged in the cooling oil. In addition, an electric field decomposition method is applied to analyze the distortion of the electric field by the structure of the conduction parts in the motor. The results indicate that the parasitic capacitances of the traction motor are increased by the influence of the cooling oil resulting in an increase in the shaft voltage.

**Keywords:** shaft voltage; parasitic capacitance; traction motor; direct-oil-cooling system; electric discharge machining; variable frequency drive; pulse width modulation; common-mode voltage

## 1. Introduction

Electric motors installed in electric vehicle propulsion systems primarily comprise of a permanent magnet synchronous motor (PMSM) and an induction motor (IM) [1]. PMSMs with high torque and power density employ permanent magnets (PMs) witha high energy density [2,3]. By contrast, IMs are more suitable for highly mature techniques and low-cost applications owing to the low yield and high price of PMs [4]. Fast switching inverters with pulse width modulation (PWM) control are widely used in electric propulsion systems to achieve variable speed control [5,6]. The inverter used to convert the DC voltage of the battery to an AC voltage causes shaft voltage and current problems [7,8]. High-frequency common-mode voltage (CMV) is induced by the high frequency switching rate of the inverters and coupled with the capacitive components of the traction motors. The induced shaft voltage causes electrical failures in sensitive bearings in traction motors and also causes electromagnetic and radio frequency interference problems [9,10].

CMV generated between motor neutral and the stator core has a high dv/dt value and can drive the capacitive currents to flow through parasitic paths [11]. However, parasitic capacitance exists between conductive parts that are insulated from each other in traction motors. Eventually, this current flows through the ball bearing to the ground and causes electric discharges, which are highly detrimental to the reliability of the bearing system. This is a severe problem because bearing failure affects motor air-gap maintenance. Therefore, parasitic capacitance must be investigated to mitigate shaft voltage in electric motors.

Several methods have been developed to analyze the parasitic capacitances in electric motors [11–14]. However, the fringing effect of the electric field has not been adequately considered in the calculation of parasitic capacitances. This can introduce significant

errors in the capacitance calculations. Generally, the electric field decomposition (EFD) method is adopted to consider the fringing effect of an electric field; in this method, the total capacitance is decomposed into three basic elements based on the partitioning of the electric field [15]. The total capacitance among the conductors can then be conveniently derived as the sum of the three elements, namely plate, terminal, and fringing capacitance [16]. Therefore, this study calculates the parasitic capacitance of the conductor parts inside the motor using the EFD method.

Meanwhile, systems with effective cooling performance should be utilized in traction motors to satisfy the requirements of high performance and power density. In this study, a direct-oil system was applied to a traction motor to achieve improved cooling performance, and an automatic transmission fluid (ATF) was used. This ATF oil directly contacts the internals and the hottest parts of the electric motor, such as the windings of the stator and rotor [17,18]. ATF oil falls under the influence of gravity in the motor, and the lower part of the motor is submerged depending on the amount of ATF oil, as shown in Figure 1. Therefore, the parasitic capacitances of traction motors with a direct-oil-cooling system are affected by the dielectric constant of the cooling oil. Several studies have investigated the parasitic capacitance and shaft voltage in various motors without a direct-oil system [19,20]. However, in previous studies on shaft voltage analysis of electric motors, shaft voltage and parasitic capacitance have not been adequately analyzed for traction motors with direct-oil-cooling systems. The relative permittivity of the cooling oil is greater than that of air can cause an increase in the parasitic capacitance, thereby causing an increase in the shaft voltage of the traction motor.

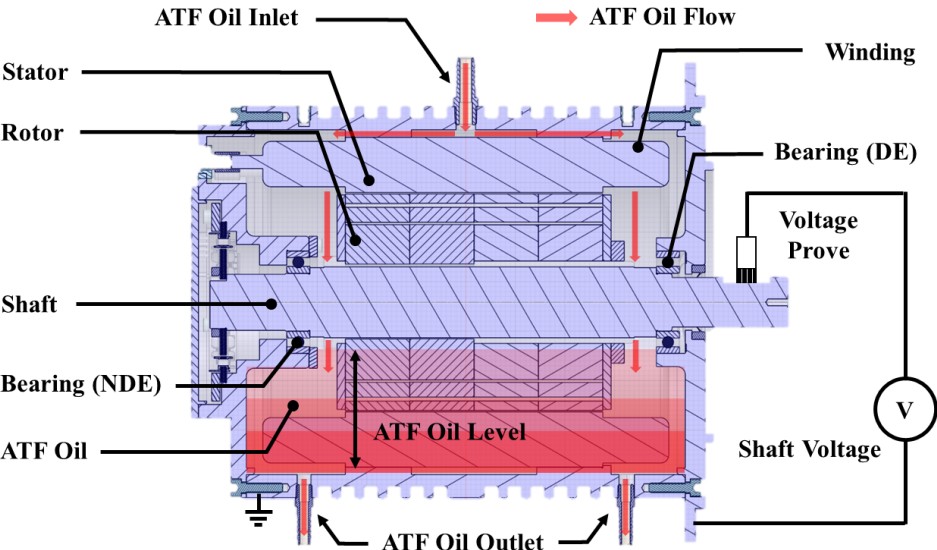

**Figure 1.** Schematic of a traction motor with direct-oil-cooling system.

Therefore, this study proposes an analytical method to determine the parasitic capacitance based on the effect of cooling oil in a direct-oil-cooling system on the shaft voltage. In addition, the parasitic capacitances calculation method inside the traction motor is proposed using the EFD Method and equivalent circuit methods for the fringing effect and charge sharing effect of conductive parts.

## 2. Parasitic Capacitance and Shaft Voltage

The parasitic capacitance of the traction motor is an unavoidable component between the conduction parts owing to their proximity to each other. The primary capacitance of the parallel plate structure, comprising of conductive plates with a cross-sectional area $A$ and distance $d$ between the plates, can be calculated as

$$C = \varepsilon_0 \varepsilon_r \frac{A}{d}. \tag{1}$$

It can be observed from the above equation that the capacitance depends on the structural characteristics of the traction motor and the characteristics of the insulating material in between. Therefore, unlike traction motors that do not have a cooling system or a cooling system outside the housing, the dielectric constant of the cooling oil must be considered in traction motors with a direct-oil-cooling system. Typically hydrocarbon ATF oils in direct-oil-cooling systems have higher relative permittivity (2.1–2.4) than air [21]. Consequently, the parasitic capacitance equivalent circuit in the traction motor changes owing to the influence of ATF oil relative permittivity.

Parasitic capacitances occur between conductors and are influenced by the shape and relative permittivity of an insulation medium between conductors. The main parasitic capacitances in the traction motors are winding-to-stator capacitance $C_{ws}$, winding-to-rotor capacitance $C_{wr}$, stator-to-rotor capacitance $C_{sr}$, and bearing capacitance $C_b$, as depicted in Figure 2a. The common-mode voltage $V_{cmv}$ at the traction motor is mirrored over the bearing by this parasitic capacitance path, causing shaft voltage $V_{sh}$ in Figure 2b.

$$V_{sh} = \frac{C_{wr}}{C_{wr} + C_{sr} + 2C_b} V_{cmv} = BVR \cdot V_{cmv} \tag{2}$$

where $BVR$ is the bearing voltage ratio of the shaft voltage to the common-mode voltage as a capacitive voltage divider [22].

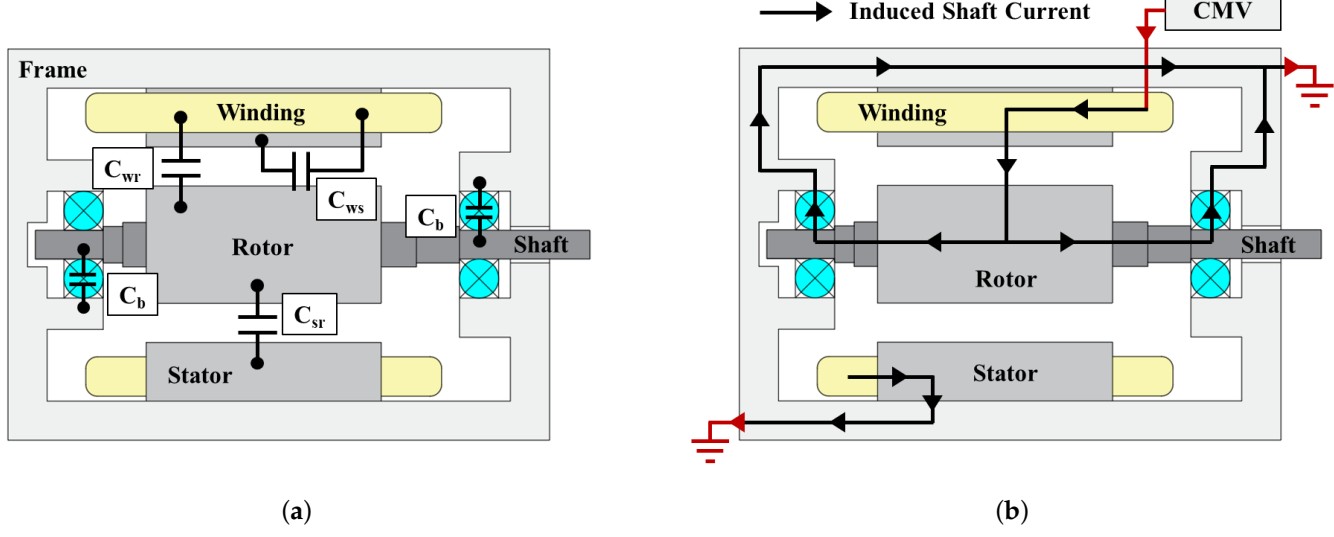

**(a)**　　　　　　　　　　　　　　　　　　　　　　　**(b)**

**Figure 2.** Parasitic capacitance and shaft current. (**a**) Main parasitic capacitances of an electric motor. (**b**) Path of shaft current induced by the common-mode voltage.

### 3. Calculation of Parasitic Capacitance Using Electric Field Decomposition Method

*3.1. Capacitance between Winding and Stator*

The winding-to-stator capacitance is the capacitance that occurs between the winding and stator. Insulation paper and windings were placed inside a practical slot, as shown in Figure 3a. The calculation of winding-to-stator capacitance should consider the insulation paper and the air layer between the stator and windings. However, it is difficult to calculate the capacitance with an air layer between non-uniform coil windings. Therefore, the air layer thickness was calculated by considering the slot fill factor (SFF) to ensure uniform air layer thickness inside the slot, as shown in Figure 3b.

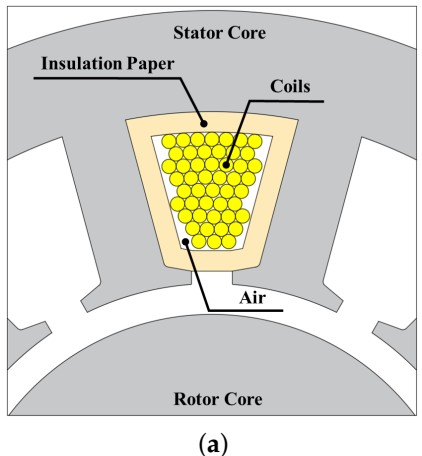
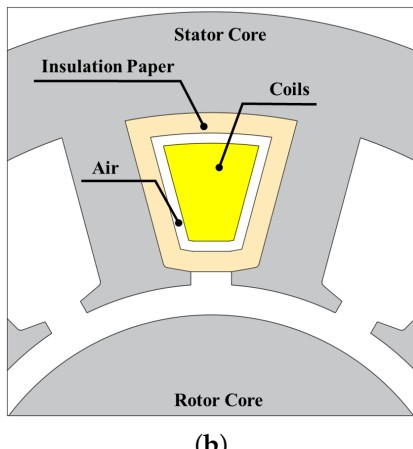

**(a)**           **(b)**

**Figure 3.** Reconstruction of the internal slot structure to calculate the parasitic capacitance. (**a**) Practical slot structure. (**b**) Internal slot structure with a uniform air layer.

As the air layer became uniform, equivalent circuits of capacitance were constructed as shown in Figure 4. A series connection configuration of air layer capacitance and insulation paper layer ca-pacitance is typically used in a winding-to-stator capacitance equivalent circuit without a direct-oil-cooling system, as shown in Figure 4a. Therefore, the capacitance of the air layer between windings and the stator ($C_{air}^{ws}$) and that of the insulation paper layer ($C_{insul}^{ws}$) are calculated as follows:

$$C_{air}^{ws} = \varepsilon_0 \varepsilon_{air} \frac{w_{us} + 2w_{sl}}{L_{wi}} L_{stk} \tag{3}$$

$$C_{insul}^{ws} = \varepsilon_0 \varepsilon_{insul} \frac{w_{us} + 2w_{sl}}{L_{ip}} L_{stk} \tag{4}$$

where $\varepsilon_0$ is the vacuum permittivity, $\varepsilon_{air}$ is the relative permittivity of air, $\varepsilon_{insul}$ is the relative permittivity of insulation paper, $w_{us}$ is the upper width of slot, $w_{sl}$ is the length of slot, $L_{wi}$ is the thickness of air layer, $L_{ip}$ is the thickness of insulation paper and $L_{stk}$ is the stack length, as shown in Figure 5. In addition, the capacitance of the insulation paper remains constant, regardless of the presence of a direct-oil-cooling system.

$$C_{insul}^{ws} = C_{insul}^{ws\_fs} = C_{insul}^{ws\_nfs} \tag{5}$$

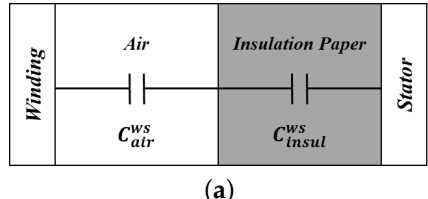

**(a)**

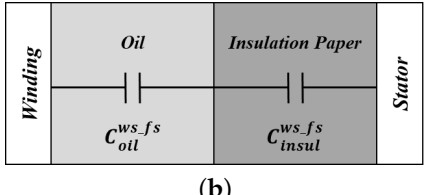

**(b)**

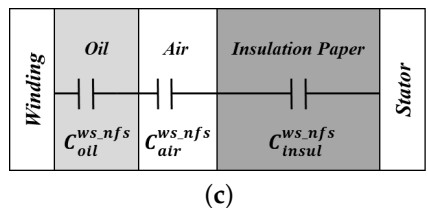

**(c)**

**Figure 4.** Winding-to-stator capacitance circuits. (**a**) Slot without cooling oil. (**b**) Oil-filled slot. (**c**) Oil-unfilled slot.

Because the two capacitances are electrically connected in series, the total capacitance without a direct-oil-cooling system ($C_{ws}$) is calculated as

$$
\begin{aligned}
C_{ws} &= \sum_{n=1}^{N_s} C_n^{ws} = \sum_{n=1}^{N_s} (C_{air}^{ws} \parallel C_{insul}^{ws}) \\
&= \sum_{n=1}^{N_s} \varepsilon_0 (w_{us} + 2w_{sl}) \frac{\varepsilon_{air}\varepsilon_{insul}}{\varepsilon_{air} L_{ip} + \varepsilon_{insul} L_{wi}} L_{stk}
\end{aligned}
\tag{6}
$$

where $C_n^{ws}$ is the winding-to-stator capacitance per slot and $N_s$ is the number of slots in the electric motor.

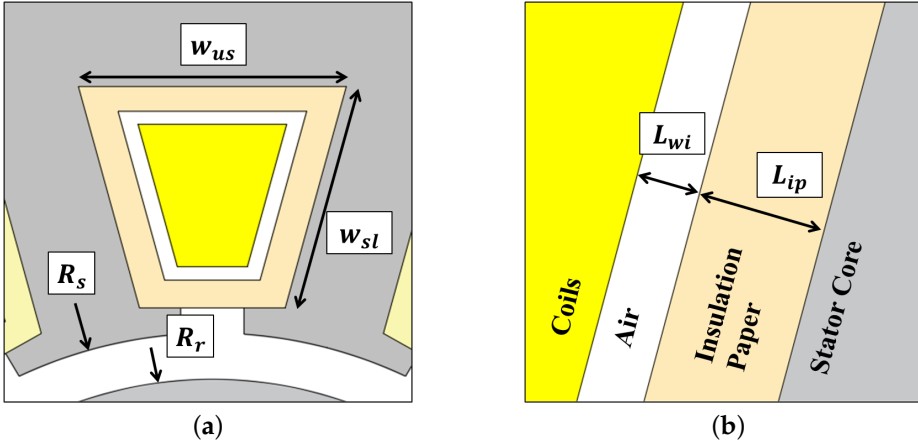

**Figure 5.** Design variables for winding-to-stator: (**a**) slot structure, and (**b**) side view of the slot.

In the direct-oil-cooling system, the slots of the electric motor are classified into oil-filled and oil-unfilled slots because a portion of the stator is submerged in cooling oil. Depending on the cooling oil level, there are situations in which the slot of the motor is submerged in the cooling oil and vice versa. The equivalent circuits of oil-filled and oil-unfilled slots are shown in Figure 4b,c, respectively.

First, in the circuit of the oil-filled slot, the air layer between the inner winding and the insulation paper was replaced by cooling oil. Therefore, because the air layer of the slot is filled with cooling oil, the calculation of the oil-layer capacitance of the oil-filled slot ($C_{oil}^{ws\_fs}$) is derived from Equation (3) as follows:

$$
\begin{aligned}
C_{oil}^{ws\_fs} &= \varepsilon_0 \varepsilon_{oil} \frac{w_{us} + 2w_{sl}}{L_{wi}} L_{stk} \\
&= \frac{\varepsilon_{oil}}{\varepsilon_{air}} C_{air}^{ws}
\end{aligned}
\tag{7}
$$

where, $\varepsilon_{oil}$ is the relative permittivity of the cooling oil. Furthermore, the capacitance per oil-fiiled slot ($C_n^{ws\_fs}$) is derived as

$$
C_n^{ws\_fs} = \varepsilon_0 (w_{us} + 2w_{sl}) \frac{\varepsilon_{oil}\varepsilon_{insul}}{\varepsilon_{oil} L_{ip} + \varepsilon_{insul} L_{wi}} L_{stk}.
\tag{8}
$$

Second, in the oil-unfilled slot, the air layer in Figure 4a is divided into a cooling oil film layer and an air layer, as shown in Figure 4c, because the cooling oil forms a thin film layer around the winding. In direct-oil-cooling systems, a liquid film of variable thickness

is formed on the conduction parts of electric motors [23]. Therefore, assuming that an oil film is formed on a cylindrical wall, the thickness of the oil film can be calculated as follows:

$$d_{film} = \sqrt[3]{\frac{3\mu Q}{2\rho g \pi R}} \tag{9}$$

where $\mu$, $\rho$ and $Q$ are the viscosity, density and volumetric flow rate of the cooling oil, respectively, and $R$ is the rotor radius [24]. Then the series capacitance of the cooling oil film and air layers is calculated as follows:

$$C_{oil}^{ws\_nfs} \parallel C_{air}^{ws\_nfs} = \varepsilon_0(w_{us} + 2w_{sl})\frac{\varepsilon_{oil}\varepsilon_{air}}{\varepsilon_{oil}(L_{ip} - d_{film}) + \varepsilon_{air}d_{film}}L_{stk}. \tag{10}$$

Because the aforementioned series capacitance is also a series configuration with insulation paper capacitance, the capacitance per oil-unfilled slot ($C_n^{ws\_nfs}$) is calculated as

$$
\begin{aligned}
C_n^{ws\_nfs} &= C_{oil}^{ws\_nfs} \parallel C_{wi\_air}^{ws\_nfs} \parallel C_{insul}^{ws\_nfs} \\
&= \varepsilon_0(w_{us} + 2w_{sl})\frac{\varepsilon_{oil}\varepsilon_{air}\varepsilon_{insul}}{\varepsilon_{oil}\varepsilon_{air}L_{ip} + \varepsilon_{air}\varepsilon_{insul}d_{film} + \varepsilon_{insul}\varepsilon_{oil}(L_{wi} - d_{film})}L_{stk}.
\end{aligned}
\tag{11}
$$

The parallel connections of the capacitance for the oil-filled slots in Equation (8) and oil-unfilled slots in Equation (11) are connected in a parallel connection configuration. Finally, when the number of oil-filled slots is $N_{fs}$, the winding-to-stator capacitance considering the cooling oil is calculated as follows:

$$C_{ws\_oil} = \sum_{n=1}^{N_{fs}} C_n^{ws\_fs} + \sum_{n=1}^{N_s - N_{fs}} C_n^{ws\_nfs}. \tag{12}$$

### 3.2. Capacitance between Winding and Rotor

The winding-to-rotor capacitance is the capacitance that occurs between the winding and rotor. An insulation paper exists between the winding and rotor, and air layers exist above and below the insulation paper as shown in Figure 6a. Therefore, the capacitance of each layer can be modeled as an equivalent circuit with a series connection, as shown in Figure 7a.

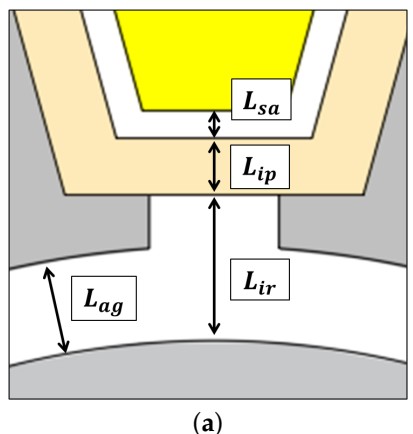

(**a**)

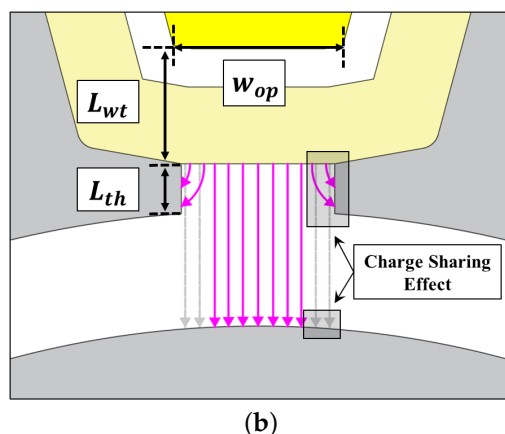

(**b**)

**Figure 6.** Design variables and charge sharing effect for winding-to-rotor capacitance (**a**) Design variables. (**b**) Charge sharing effect of the slot structure.

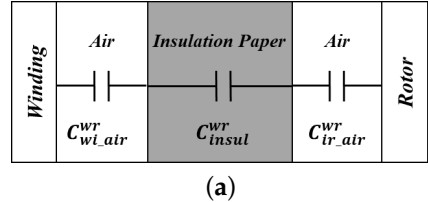

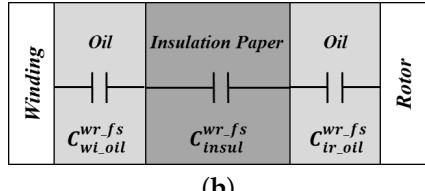

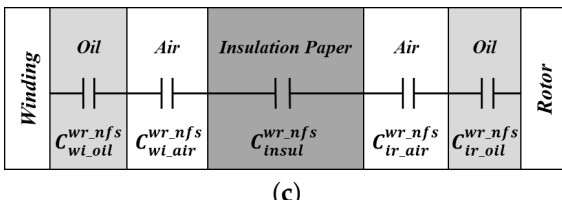

**Figure 7.** Winding-to-stator capacitance circuits: (**a**) without cooling oil, (**b**) oil-filled slot, and (**c**) oil-unfilled slot.

Using the air layer capacitance above the insulation paper ($C^{wr}_{wi\_air}$), the insulation paper capacitance ($C^{wr}_{insul}$), and capacitance below the insulation paper ($C^{wr}_{ir\_air}$), winding-to-rotor capacitance without cooling oil ($C^{wr}_n$) is calculated as

$$C^{wr}_n = C^{wr}_{wi\_air} \parallel C^{wr}_{insul} \parallel C^{wr}_{ir\_air}$$
$$= \varepsilon_0 \frac{\varepsilon_{air}{}^2 \varepsilon_{insul} w_{op}}{\varepsilon_{air} \varepsilon_{insul}(L_{ir} + L_{sa}) + \varepsilon_{air}{}^2 L_{ip}} L_{stk} \tag{13}$$

where $L_{sa}$ is the distance between the windings and insulation paper, $L_{ip}$ is the thickness of the insulation paper, $L_{ir}$ is the distance between the insulation paper and the rotor, $w_{op}$ is the bottom width of the slot.

However, the electric field formed between the winding and rotor undergoes a charge-sharing effect as shown in Figure 6b. The charge sharing effect is a phenomenon in which the field from one conductor can be shared by two or more wire surfaces [16]. The capacitance $C_1'$ affected by the charge sharing effect of $C_2$ is calculated as follows:

$$C_1' = f_{cs}(C_1, C_2) = C_1 \cdot \frac{C_1}{C_1 + C_2} \tag{14}$$

where $C_1$, $C_2$ are capacitances between two electrodes without considering charge sharing, $f_{cs}(C_1, C_2)$ is the charge sharing effect function [16].

Therefore the effective area of the electric field is reduced owing to the charge sharing effect because the electric field from the winding is coupled to both the rotor and stator. The winding-to-rotor capacitance considering the charge sharing effect is calculated using the following equation:

$$C^{wr\prime}_n = f_{cs}(C^{wr}_n, C^{wr}_{ws\_fringing\_air}) = C^{wr}_n \cdot \frac{C^{wr}_n}{C^{wr}_n + C^{wr}_{ws\_fringing\_air}} \tag{15}$$

where $C^{wr}_{ws\_fringing\_air}$ is the capacitance resulting from the electric field induced within winding from the sidewall of the stator teeth cap. The electric field of $C^{wr}_{ws\_fringing\_air}$ can be calculated based on the fundamental components of fringe capacitance and terminal capacitance.

$$C^{wr}_{ws\_fringing\_air} = 2\left( \varepsilon_0 \varepsilon_{air} \frac{4}{\pi} \ln 2 + \int_{L_{wt}}^{L_{wt}+L_{th}} \varepsilon_0 \varepsilon_{air} \frac{2}{\pi x} dx \right) L_{stk}$$
$$= 2\left( \varepsilon_0 \varepsilon_{air} \frac{4}{\pi} \ln 2 + \varepsilon_0 \varepsilon_{air} \frac{2}{\pi} \ln(1 + \frac{L_{th}}{L_{wt}}) \right) L_{stk} \tag{16}$$

where $L_{th}$ is the distance between the winding and teeth and $L_{wt}$ is the thickness of the stator teeth cap.

The winding-to-rotor capacitance considering the direct-oil-cooling system can be divided into oil-filled and oil-unfilled slot, similar to the winding-to-stator capacitance. The winding-to-rotor capacitance of the oil-filled slot $C_n^{wr\_fs}$ can be calculated by modifying Equation (13), as follows.

$$C_n^{wr\_fs} = \varepsilon_0 \frac{\varepsilon_{oil}^2 \varepsilon_{insul} w_{op}}{\varepsilon_{oil} \varepsilon_{insul}(L_{ir} + L_{sa}) + \varepsilon_{oil}^2 L_{ip}} L_{stk}. \tag{17}$$

Similarly, $C_n^{wr\_fs\prime}$ is applied using the same method as that for a slot without cooling oil, considering the charge sharing effect.

$$C_n^{wr\_fs\prime} = f_{cs}(C_n^{wr\prime}, C_{ws\_fringing\_oil}^{wr}) \tag{18}$$

where the capacitance that shares the charge between the winding and rotor in a slot submerged in cooling oil ($C_{ws\_fringing\_oil}^{wr}$) is calculated as

$$C_{ws\_fringing\_oil}^{wr} = \frac{\varepsilon_{oil}}{\varepsilon_{air}} C_{ws\_fringing\_air}^{wr}. \tag{19}$$

Moreover, the capacitance of the oil-non-filled slot with oil, considering the charge sharing effect and the thin film layer on the winding and rotor surface is calculated as follows.

$$C_n^{wr\_nfs\prime} = f_{cs}(C_n^{wr\_nfs}, C_{ws\_fringing\_air}^{wr}) \tag{20}$$

$$C_n^{wr\_nfs} = C_{wi\_oil}^{wr\_nfs} \parallel C_{wi\_air}^{wr\_nfs} \parallel C_{aa\_insul}^{wr\_nfs} \parallel C_{ir\_air}^{wr\_nfs} \parallel C_{ir\_oil}^{wr\_nfs} \tag{21}$$

Finally, the winding-to-rotor capacitance in the presence of a direct-oil-cooling system is calculated as

$$C_{wr\_oil} = \sum_{n=1}^{N_{fs}} C_n^{wr\_fs\prime} + \sum_{n=1}^{N_s - N_{fs}} C_n^{wr\_nfs\prime}. \tag{22}$$

*3.3. Capacitance between Stator and Rotor*

Stator-to-rotor capacitance occurs between the stator and rotor. In previous studies, the stator-to-rotor capacitance was calculated using cylindrical geometry, such as a coaxial cable [12,25–27]. This calculation method assume that the charge is uniformly distributed on the outer surface of the inner conductor and the inner surface of the outer conductor. However, the fringing effect represents the non-uniform electric fields around the edge of stator teeth as shown in Figure 8. Furthermore, the charges on both walls of the stator teeth can be shared via the insulation paper, such as the charge sharing effect of the winding-to-rotor capacitance.

The configuration of the capacitance equivalent circuit of the stator-to-rotor capacitance is simpler than that of other parasitic capacitances, as shown in Figure 9a. Using the angle of the stator teeth $A_{sl}$, the outer diameter of the rotor $R_r$, and the inner diameter of the stator $R_s$, the stator-to-rotor capacitance per slot can be calculated as follows:

$$\begin{aligned} C_n^{sr\prime} &= C_{air}^{sr} + C_{sharing\_air}^{sr} \\ &= \varepsilon_0 \varepsilon_{air} \frac{2\pi A_{sl}}{\ln \frac{R_s}{R_r}} L_{stk} + C_{sharing\_air}^{sr} \end{aligned} \tag{23}$$

where the first term is the capacitance without considering the fringing effect, and $C_{sharing\_air}^{sr}$ is the capacitance due to the fringing and charge sharing effect at the end of the stator teeth and is calculated using the charge sharing effect function as

$$
\begin{aligned}
C_{sharing\_air}^{sr} &= f_{cs}(C_{fringe\_air}^{sr}, C_{ws\_fringing\_air}^{sr}) \\
&= f_{cs}\left(2\varepsilon_0\varepsilon_{air}\left[\frac{4}{\pi}\ln 2 + \frac{2}{\pi}\ln\left(1 + \frac{L_{th}}{L_{ag}}\right)\right]L_{stk}, C_{ws\_fringing\_air}^{wr}\right)
\end{aligned}
\tag{24}
$$

where $L_{ag}$ is the length of the airgap, and $C_{fringe\_air}^{sr}$ from both sides of the stator teeth can be calculated as the sum of the terminal capacitance and fringing capacitance in the EFD method. The $C_{ws\_fringing\_air}^{sr}$ is the same as $C_{ws\_fringing\_air}^{wr}$ in Equation (16), because the capacitance is generated by the electric field that originates within the side walls of the stator from the winding. Therefore, the stator-to-rotor capacitance per slot in the absence of the direct-oil-cooling system is given by

$$
C_n^{sr\prime} = \varepsilon_0\varepsilon_{air}\left(\frac{2\pi A_{sl}}{\ln\frac{R_s}{R_r}} + \frac{2\left[\frac{4}{\pi}\ln 2 + \frac{2}{\pi}\ln\left(1 + \frac{L_{th}}{L_{ag}}\right)\right]^2}{\frac{4}{\pi}\ln 2 + \frac{2}{\pi}\ln\left(1 + \frac{L_{th}}{L_{wt}}\right)\left(1 + \frac{L_{th}}{L_{ag}}\right)}\right)L_{stk}
\tag{25}
$$

In the direct-oil-cooling system, motor slots are divided into oil-filled slots and oil-unfilled slots, as in other parasitic capacitance calculations.

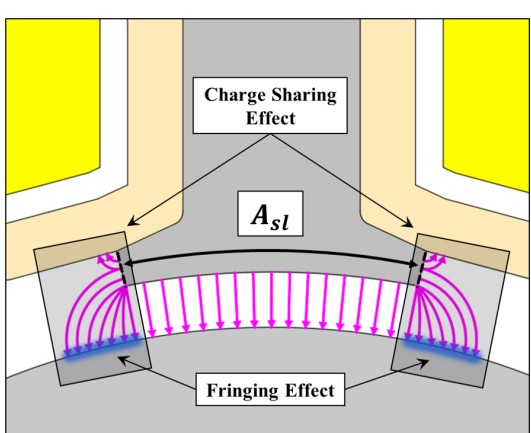

**Figure 8.** Electric field distribution between the stator and rotor for parasitic capacitance calculation.

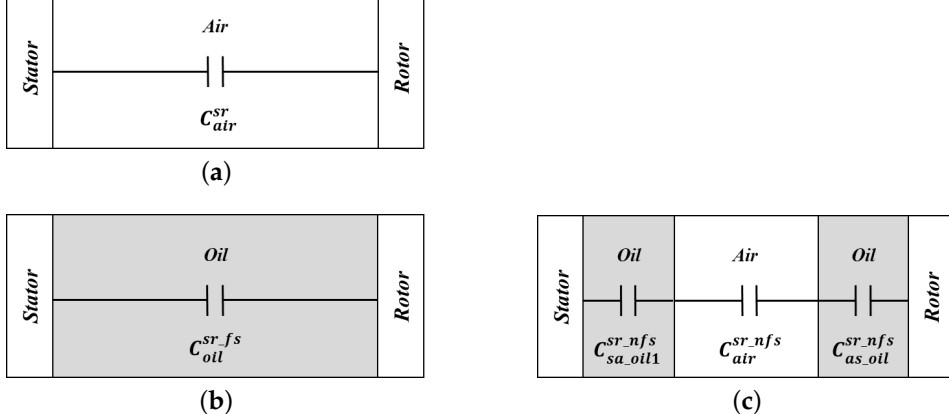

**Figure 9.** stator-to-rotor capacitance circuits: (**a**) without cooling oil, (**b**) oil-filled slot, and (**c**) oil-non-filled slot.

First, the capacitance of the oil-filled slot $C_n^{sr\_fs}$ is calculated by replacing the relative permittivity of air with oil in Equation (25).

$$C_n^{sr\_fs'} = \frac{\varepsilon_{oil}}{\varepsilon_{air}} C_n^{sr'} \tag{26}$$

Second, the capacitance with an oil-unfilled slot ($C_n^{sr\_nfs}$) is calculated by considering the oil film thickness in Equation (9), as follows.

$$C_n^{sr\_nfs} = \varepsilon_0 \varepsilon_{air} 2\pi A_{sl} \left[ \left( \ln \frac{R_r + d_{film}}{R_r} \right)^{-1} + \left( \ln \frac{R_r - d_{film}}{R_r + d_{film}} \right)^{-1} + \left( \ln \frac{R_s}{R_s - d_{film}} \right)^{-1} \right] L_{stk}$$
$$+ f_{cs} \left( C_{fringe\_air}^{sr}, C_{ws\_fringing\_air}^{sr} \right) \tag{27}$$

Finally, the capacitance with the direct-oil-cooling system is calculated as

$$C_{sr\_oil} = \sum_{n=1}^{N_{fs}} C_n^{sr\_fs'} + \sum_{n=1}^{N_s - N_{fs}} C_n^{sr\_nfs'}. \tag{28}$$

## 4. Calculations and Validation of Traction Motor with Direct-Oil-Cooling System

This section presents the analysis of the traction motor using the analytical method proposed in this study to elucidate the effect of the direct-oil-cooling system on the parasitic capacitance and shaft voltage characteristics of the electric motors. The electric propulsion motor used in the electric vehicle to analyze the shaft voltage is an interior permanent magnet synchronous motor (IPMSM) of 160 kW power rating (350 Nm at 4400 rpm) as shown in Figure 10a, and its specifications are listed in Table 1.

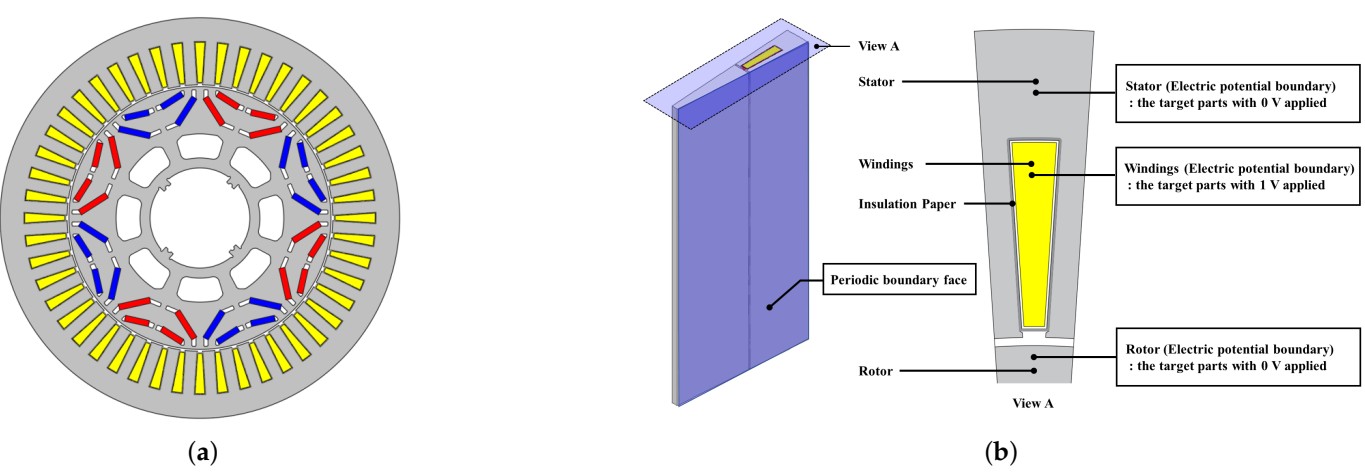

**(a)**                                                                 **(b)**

**Figure 10.** Traction motor schematic. (**a**) Motor structure. (**b**) 3-D partial model and boundary conditions for the FEM analysis.

In practical electric motors, the axial edge effect and end-turn effect of the windings cause an additional parasitic capacitance. Although it is difficult to calculate parasitic capacitance caused by the axial edge effect and the end-turn effect of windings, analytical methods have been proposed in several studies [28–30]. However, the method of calculating parasitic capacitance proposed in this study was validated by FEM analysis in the active length section of the electric motor because the proposed method focused on the fringing effect of the electric field on the active length of the electric motor and the influence of cooling oil inside the slot. The partial model is used to reduce FEM calculation costs because the geometry and electric field have periodicity, as illustrated in Figure 10b. The parasitic

capacitance is calculated from the surface charge value using the electric potential difference of 1 V between the conductive parts that have an electric potential applied to them.

The traction motor used in this study had a direct-oil-cooling system with a cooling oil level of approximately 35%. As shown in Figure 11, for ease of calculations, 18 slots were submerged in ATF oil because the cooling oil level is converted to an integer depending on the area of the slot submerged in the cooling oil. The design variables used in capacitance calculations and shaft voltage analysis are listed in Table 2.

**Table 1.** Specifications of the traction motor with direct-oil-cooling system.

| Parameter | Value | Units |
|---|---|---|
| Poles/Slots | 8/48 | - |
| Maximum Power | 160 | kW |
| Maximum Speed | 15,000 | RPM |
| DC link voltage | 600 | V |
| Switching Frequency | 8 | kHz |

**Table 2.** Design parameters of the analyzed traction motor.

| Parameter | Definition | Value | Units |
|---|---|---|---|
| $N_s$ | Number of slots | 48 | EA |
| $N_{fs}$ | Number of oil-filled slots | 18 | EA |
| $\varepsilon_0$ | Permittivity of vacuum | $8.854 \times 10^{-12}$ | F/m |
| $\varepsilon_{air}$ | Relative permittivity of air | 1.00056 | - |
| $\varepsilon_{oil}$ | Relative permittivity of cooling oil (ATF) | 2.4 | - |
| $\varepsilon_{insul}$ | Relative permittivity of insulation paper | 2.7 | - |
| $w_{us}$ | Upper Width of slot | 5.94 | mm |
| $w_{sl}$ | Length of slot | 21 | mm |
| $L_{wi}$ | Thickness of air layer | 0.25 | mm |
| $L_{ip}$ | Thickness of insulation paper | 0.25 | mm |
| $d_{film}$ | Thickness of cooling oil film | 0.036 | mm |
| $L_{sa}$ | Distance between winding and insulation paper | 0.25 | mm |
| $L_{ir}$ | Distance between insulation paper and rotor | 1.498 | mm |
| $w_{op}$ | Bottom width of slot | 3 | mm |
| $L_{th}$ | Distance between winding and teeth | 0.598 | mm |
| $L_{wt}$ | Thickness of stator teeth cap | 0.5 | mm |
| $A_{sl}$ | Angle of the stator teeth | $0.029\,\pi$ | rad |
| $R_r$ | Outer radius of rotor | 65.6 | mm |
| $R_s$ | Inner radius of stator | 66.5 | mm |
| $L_{ag}$ | Length of the air-gap | 0.9 | mm |
| $L_{stk}$ | Stack length | 158 | mm |
| $C_{b,DE}, C_{b,NDE}$ | Capacitance of the bearing | 208.87 | pF |

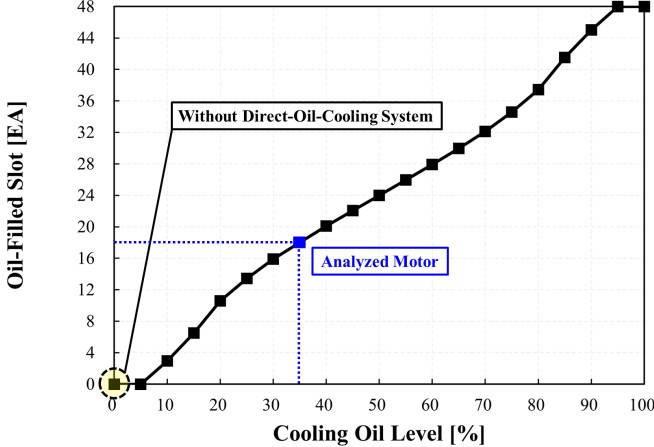

**Figure 11.** Number of slots according to the level of cooling oil.

### 4.1. Parasitic Capacitances and Bearing Voltage Ratio

This subsection describes the variations in the parasitic capacitances in the traction motor according to the cooling oil level. Figure 12 shows the results of the parasitic capacitance and BVR calculated using the proposed analytical method. In addition, the calculation results were compared with the finite element method (FEM) simulation results to validate the proposed method. Evidently, simulation results are accurately predicted. When the cooling oil is 0%, the capacitances are under conditions similar to without a direct-oil-cooling system. Therefore, the relative permittivity and film of the cooling oil are not considered. When the cooling oil level exceeds 0%, the slots are divided into oil-filled slots and oil-unfilled slots. As the oil level increases, all three parasitic capacitances increase because of the increased number of slots affected by the relative permittivity of the cooling oil.

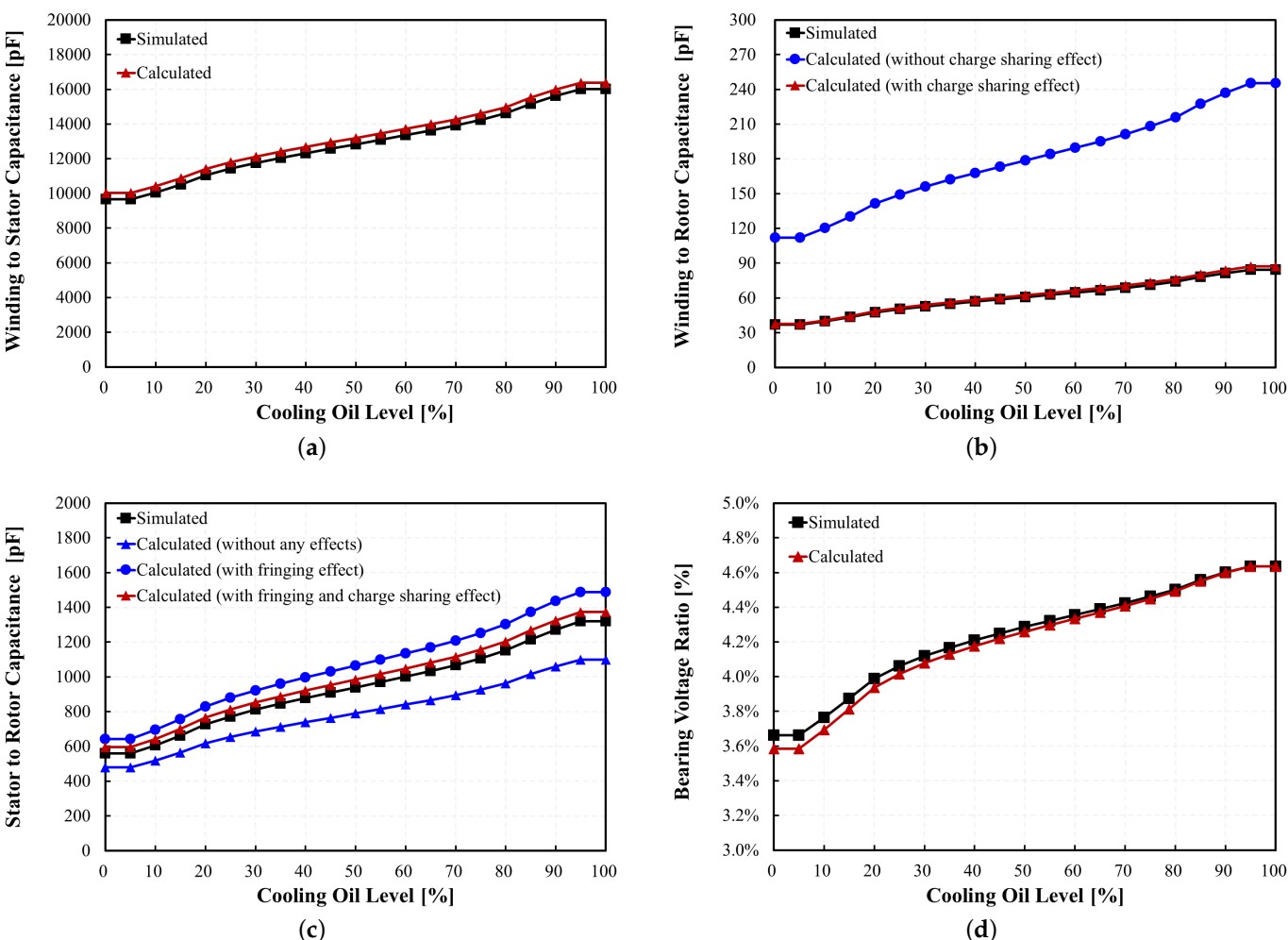

**Figure 12.** Results of calculation and simulation for parasitic capacitance and bearing voltage ratio. (**a**) Winding-to-stator capacitance; (**b**) winding-to-rotor capacitance; (**c**) stator-to-rotor capacitance; and (**d**) bearing voltage ratio.

The winding-to-stator capacitance increases as shown in Figure 12a because the cooling oil level increases, resulting in an increase in the number of oil-affected slots. The winding-to-rotor capacitance reduces the effective area for the calculation because the electric field from the winding also affects the stator teeth. This phenomenon can be addressed by considering the charge sharing effect. As shown in Figure 12b, the winding-to-rotor capacitance considering the charge sharing effect (red line and triangle marker) has a lower value than the capacitances without considering the charge sharing effect (blue line and circle marker). The stator-to-rotor capacitance is calculated by considering the charge

sharing and fringing effects, as shown in Figure 12c. The fringing effect extends the electric field between the stator and the rotor. The capacitance, considering the fringing effect (blue line and circle marker), has a higher value than that without considering the fringing effect (blue line and triangle marker). However, this value is higher than the predicted via simulation (black line and square marker) because the electric field from the stator affects only the rotor. Therefore, the result of recalculating the stator-to-rotor capacitance with the fringing effect considering the charge sharing effect yields results that are similar to the simulation results. Moreover, the BVR tended to increase as the cooling oil level increased. However, the rate of increase rate the BVR is lower than those of the parasitic capacitances because each parasitic capacitance increases, as shown in Figure 12d.

### 4.2. Shaft Voltage

Because the traction motor of the analysis model uses a PWM inverter, common-mode voltage, which is the source of shaft voltage, is generated, as shown in Figure 13a. When $V_{an}$, $V_{bn}$ and $V_{cn}$ are the phase voltages, CMV is given by

$$V_{CMV} = \frac{V_{an} + V_{bn} + V_{cn}}{3}. \tag{29}$$

Figure 14 shows the simulation results of the common-mode voltage at 360 V and 600 V when the traction motor operates at 3000 rpm. The shaft voltage is simulated by the BVR calculated using the common-mode voltage and parasitic capacitance. Figure 15 shows the simulation results of the shaft voltage according to the cooling oil level when each common-mode voltage is applied. The shaft voltage is simulated under normal conditions when the traction motor has an oil level of 35% and under the assumed condition of no direct-oil-cooling system. The shaft voltage increases when the cooling oil level is considered for both the 360 V and 600 V systems owing to the increased BVR. The peak-to-peak voltage of shaft voltage with 35% cooling oil level is 15.24 V while shaft voltage without cooling oil is 13.26 V. Therefore, the shaft voltage considering the cooling oil level in the analyzed motor is found to be 15% more than that without cooling oil.

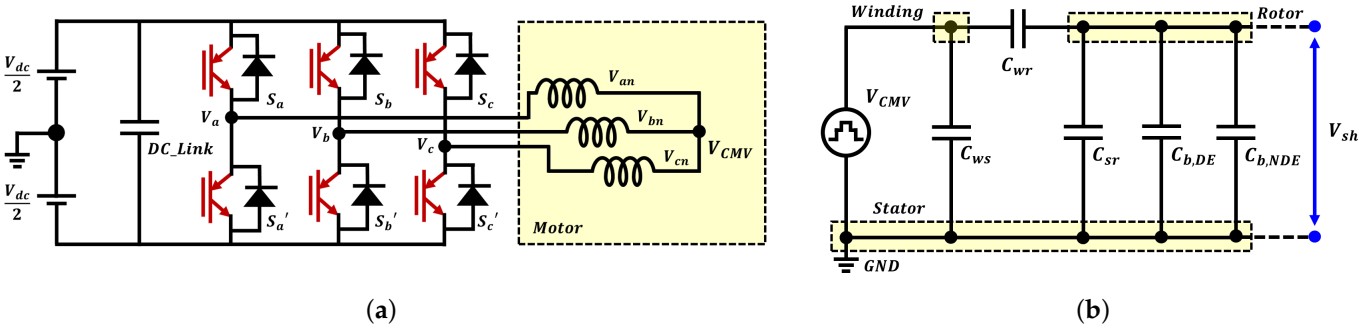

**Figure 13.** Circuit configuration of driver system and parasitic capacitance. (**a**) Typical PWM inverter; (**b**) parasitic capacitance circuit for shaft voltage analysis.

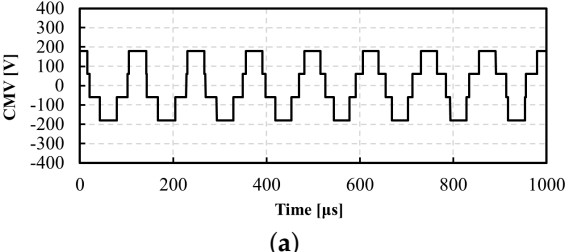

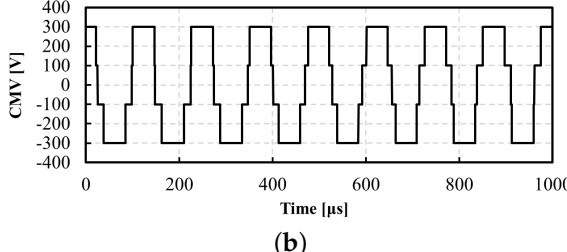

**Figure 14.** Simulated common-mode voltage waveform at (**a**) 360 V and (**b**) 600 V.

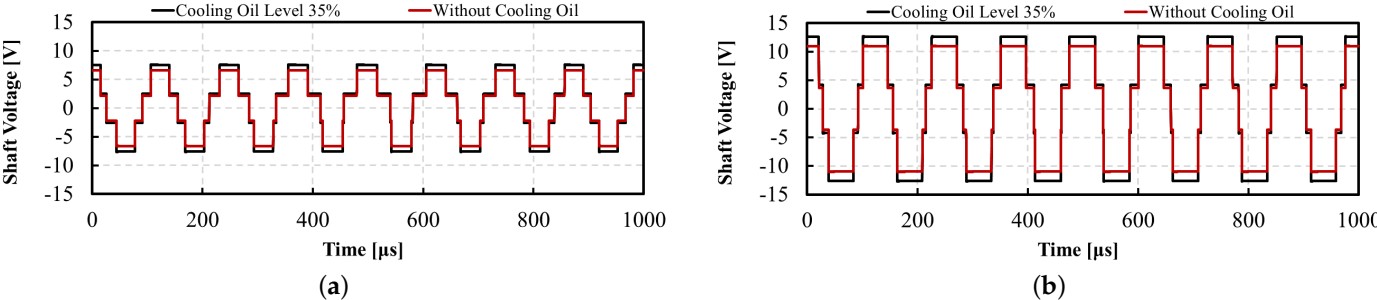

(a)

(b)

**Figure 15.** Simulated shaft voltage waveform at (**a**) 360 V and (**b**) 600 V.

## 5. Experimental Results

In an actual traction motor, electric discharge machining (EDM) electrostatic discharge occurs inside the bearing when the shaft voltage depicted in Figure 15 exceeds the bearing lubrication grease insulation resistance. Some researchers have developed a statistical model using a probability density function because it is difficult to formulate an exact mathematical expression for the EDM electrostatic discharge [31,32]. As shown in Figure 16, the simulation model is configured to analyze the discharged shaft voltage using the lumped network method in [32].

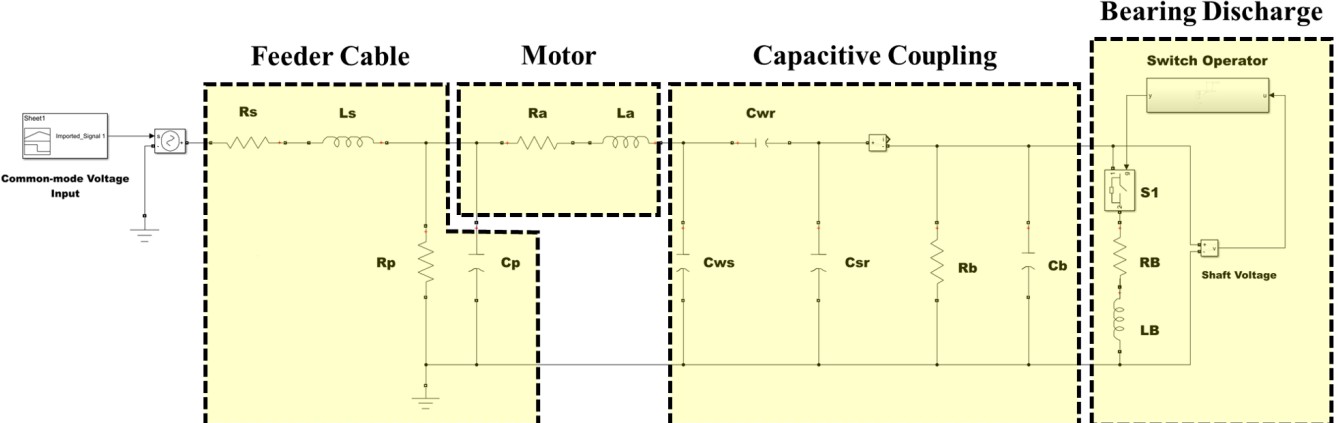

**Figure 16.** Simulation model for shaft voltage discharge in bearing.

The simulation model for the bearing discharge behavior is modeled by combining the feeder cable, high-frequency equivalent impedance for the electric motor, capacitive coupling, and bearing discharge circuit as shown in Figure 16. In the feeder cable circuit, the equivalent series resistance and inductance are denoted by $R_s$ and $L_s$, respectively, while the equivalent transversal capacitance and resistance are given by $C_p$ and $R_p$, respectively. The series equivalent impedance of the motor are given by $R_a$ and $L_a$. The capacitive coupling circuit has the same parameters as the simulation model shown in Figure 13b, and $R_b$ indicates the insulation resistance of the bearing. The bearing discharge circuit is modeled as a switch with internal inductance $LB$ and resistance $RB$ [33]. When the voltage across the bearing reaches a given threshold, the switch operator block simulates the discharge of the bearing by operating switch $S1$.

Figure 17 displays images of the experimental setup of the traction motor with a direct-oil-cooling system. The traction motor under the experiment is identical to the electric motor simulated in the previous section. To be more specific, ATF oil is injected into the upper cooling oil inlet of the traction motor, which then flows to the two cooling oil outlets at the bottom of the traction motor. Dedicated equipment, such as a brush-type voltage probe, is required to measure the voltage generated on the rotating shaft of the traction motor.

The shaft voltage of the traction motor was measured when the traction motor was operated at 3000 rpm in the 360 V and 600 V systems. As shown in Figure 17b, the differential voltage probe contacts the rotor shaft with a brush to measure the shaft voltage. The shaft voltage measurement and simulation data are presented in Figure 18. A good agreement between the simulated and measured results can be observed for discharge voltage level and pattern. To comprehensively compare the change in discharge voltage according to DC link voltage, the histograms of the probability density for discharge voltages are shown in Figure 19. The discharge voltage is effectively described using an inverse Gaussian distribution [31]. Moreover, the bearing discharge phenomenon occurs at higher voltage levels, specifically when the DC link voltage is 600 V. These results indicate a high probability of bearing failure in high DC link voltage systems.

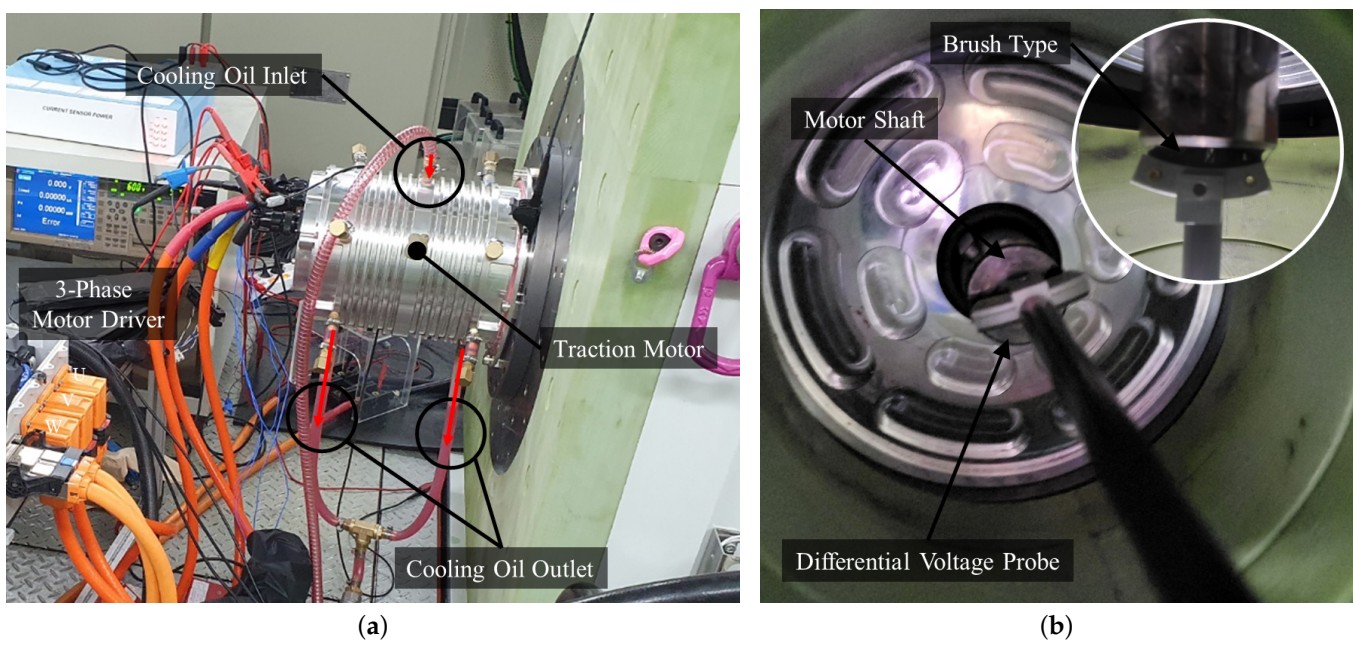

**Figure 17.** Photographs of the experimental setup. (**a**) Traction motor; (**b**) Shaft voltage measurement.

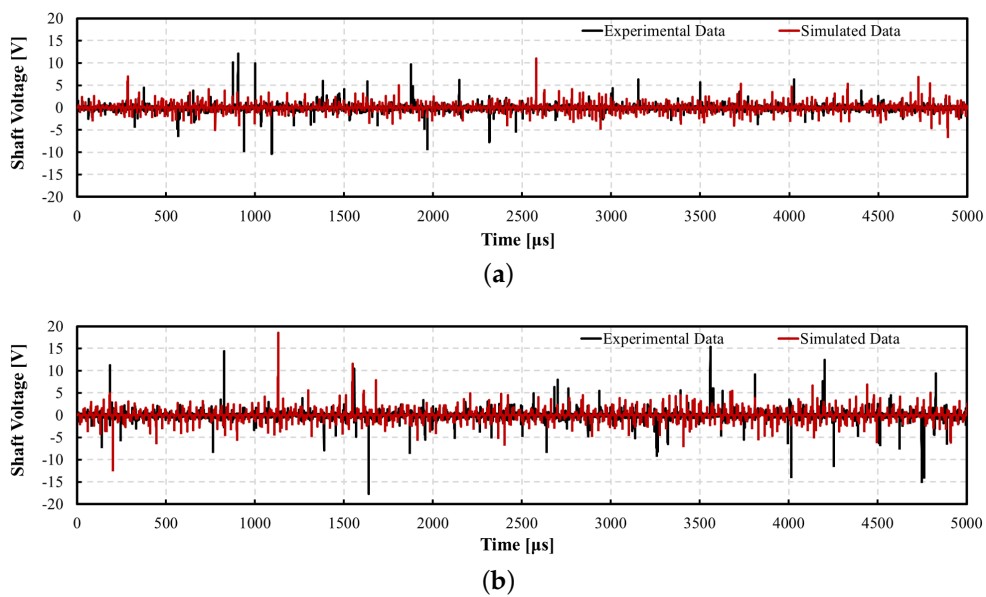

**Figure 18.** Shaft voltage obtained from experiments and bearing discharge model at 3000 rpm: (**a**) 360 V; (**b**) 600 V.

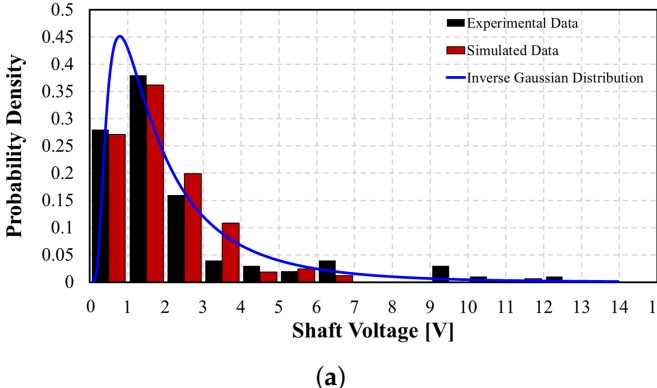
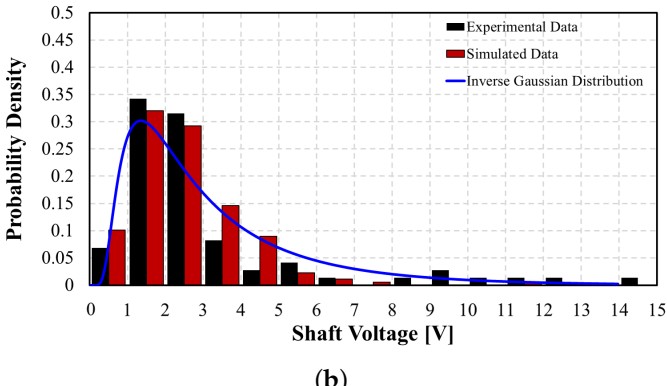

(**a**)　　　　　　　　　　　　　　　　　　(**b**)

**Figure 19.** Histogram and inverse Gaussian distribution for shaft voltage at 3000 rpm: (**a**) 360 V; (**b**) 600 V.

## 6. Conclusions

In previous studies related to shaft voltage analysis of electric motors, direct-oil-cooling systems of a traction motor were not considered for shaft voltage and parasitic capacitance analysis. Furthemore, studies that consider the distortion of the electric field, such as the fringing effect, in relation to the calculation of parasitic capacitances in electric motors remain limited. Therefore, this study developed an analytical method for the parasitic capacitance and shaft voltage of a traction motor with a direct-oil-cooling system using the EFD method.

The equivalent capacitance circuits were configured based on whether the slot is submerged in the cooling oil. Subsequently, the parasitic capacitances inside the traction motor were calculated using the EFD Method and equivalent circuit methods. In particular, the winding-to-rotor capacitance and stator-to-rotor capacitance were significantly influenced by the electrical sharing and fringing effects. The winding-to-rotor exhibited a charge sharing effect, sharing an electric charge with the stator teeth, and the stator-to-rotor capacitance exhibited both a fringing effect and a charge sharing effect.

In addition, as the oil level increased, all three parasitic capacitances increased because of the increased number of slots affected by the relative permittivity of the cooling oil. The calculation results were compared with those obtained via FEM simulation to validate the proposed analytical method; it was found that the predicted and simulation results showed good agreement. Therefore, it can be deemed that the proposed EFD-based analytical method is suitable for determining the distortion of the electric field generated inside the slot of the traction motor. Overall, the results of this study indicate that the parasitic capacitances of the traction motor are increased because of the higher relative permittivity of the cooling oil relative to that of air, resulting in an increase in the shaft voltage.

**Author Contributions:** Conceptualization, C.-H.K. and S.-B.J.; data curation, H.-J.Y. and N.-H.K.; investigation and Writing-original draft, C.-H.K.; validation, H.-C.J.; resources, R.-E.K.; writing—review and editing, H.-C.J. and S.-Y.J.; project administration, S.-Y.J. All authors have read and agreed to the published version of the manuscript.

**Funding:** This work was supported by the Technology Innovation Program (or Industrial Strategic Technology Development Pro-gram) (20010437, Development of high performance electric drive system technologies for xEV capable of realizing more than 3.5 kW/kg power density) funded By the Ministry of Trade, Industry Energy (MOTIE, Korea).

**Institutional Review Board Statement:** Not applicable.

**Informed Consent Statement:** Not applicable.

**Data Availability Statement:** Not applicable.

**Conflicts of Interest:** The authors declare no conflict of interest.

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
