# Peer review of "Calculation of Parasitic Capacitance to Analyze Shaft Voltage of Electric Motor with Direct-Oil-Cooling System"

_processes, doi:10.3390/pr10081541_

Round 1

Reviewer 1 Report

The article under review presents the results of studies of the analytical method proposed by the authors for determining parasitic capacitance in oil-cooled traction motors. This task is especially relevant in electric vehicle drives using AC motors powered by an inverter with pulse width modulation.

In the Introduction and literature review, the prerequisites for conducting research are considered in sufficient detail, and the purpose of the paper is formulated. In the main parts of the paper, the problem of parasitic capacitance and shaft voltage is considered in detail, a method for calculating parasitic capacitance is proposed (taking into account the capacitance between the winding and the stator, the winding and the rotor, the stator and the rotor). The calculation and validation for a real permanent magnet synchronous motor with the direct-oil-cooling system of 160 kW power rating, used in an electric vehicle, is given. The results of experimental studies are presented too.

In general, this is an interesting work, the results of which may be useful to specialists in the field of electric motors in application to electric vehicles.

During the review, I would like to ask the authors for a few clarifications:

  1. What are the parameters of the motor used in the experimental setup? Is this the same motor for which the calculations were made in Section 4?
  2. It seems that a more detailed explanation of the conclusion made by the authors about the correspondence between the simulated and measured results is necessary.

I congratulate the authors on a job well done and recommend the paper for acceptance after minor revision.

Author Response

We would like to thank you for spending your precious time on providing insightful comments, which have greatly helped us to improve the quality of our manuscript. We have revised and improved our paper according to your suggestions. 

Author Response

We would like to thank you for spending your precious time to provide insightful comments, which have greatly helped us improve the quality of our manuscript. We have improved our paper according to your suggestions and made a response sheet.

Round 2

Reviewer 2 Report

Thank you for having considered all the comments stated by the referee. The manuscript looks much better now.